Longitudinal relations between non-suicidal self-injury and both depression and anxiety among senior high school adolescents: a cross-lagged panel network analysis

Zhao Haiyan
Zhou Aibao zhoulabpapers@163.com
Department of Psychology, The Northwest Normal University , Lanzhou , Gansu province , China
Zhong Bao-Liang
Electronic publication date: 2024 Oct 7
Publication date: 2024
Volume: 12
Electronic Location ID: e18134
Received 2024 May 8; Accepted 2024 Aug 29
Copyright: ©2024 Zhao and Zhou
Copyright year: 2024
Copyright holder: Zhao and Zhou
License: This is an open access article distributed under the terms of the Creative Commons Attribution License, which permits unrestricted use, distribution, reproduction and adaptation in any medium and for any purpose provided that it is properly attributed. For attribution, the original author(s), title, publication source (PeerJ) and either DOI or URL of the article must be cited.
License URL: https://creativecommons.org/licenses/by/4.0/

Keywords: Non-suicidal self-injury, Depression, Anxiety, Comorbidity, Gender difference

Funding: The research fund from the Natural Science Foundation of China 32160202 32360202 This work is supported by the research fund from the Natural Science Foundation of China under grant No. 32160202 and 32360202. The funders had no role in study design, data collection and analysis, decision to publish, or preparation of the manuscript.

==============================
Background

Comorbidity between non-suicidal self-injury (NSSI) and depression and anxiety was common. In the framework of network theory, the examination of directionality and gender differences in longitudinal relationships at the symptom level made a significant contribution to the understanding of comorbidity. Therefore, this study employed cross-lagged panel network analysis to investigate the longitudinal interrelations between NSSI and depression and anxiety in Chinese adolescents, with a focus on gender differences.

Method

The study was conducted with a sample of 884 senior high school students (F/M: 481/403; mean age: 15.19 ± 0.48 years) from Jinchang City, Gansu Province, China. All respondents completed the Adolescent Non-Suicidal Self-Injury Assessment Questionnaire and the two subscales (depression and anxiety) of the Brief Symptom Inventory at two intervals. The data were estimated in R 4.2.0 to construct the cross-lagged panel network (CLPN).

Results

The CLPN results uncovered the gender differences. For boys, self-hitting and feeling scared emerged as central symptoms. Cutting predicted subsequent feelings of sadness (β =  − 0.57), tension (β =  − 0.52) and indifference (β =  − 0.49), potentially serving as a bridge connecting NSSI to depression and anxiety. For girls, biting themselves and feeling scared were central symptoms. Carving and skin rubbing predicted subsequent feelings of indifference (β =  − 0.31, −0.21), bridging NSSI to depression and anxiety. In addition, feeling scared emerged as the key bridge symptom connecting depression and anxiety.

Conclusion

The findings showed the gender-specific developmental characteristics of the directional relations between NSSI and depression and anxiety at the symptom level. They provided new insights into the comorbidity of NSSI and depression and anxiety, carrying important implications for the screening and intervention of adolescent NSSI.

Introduction

Non-suicidal self-injury (NSSI) is defined as the direct and deliberate destruction of one’s body tissues without suicidal intent (Nock, 2009). It has become a significant global public health concern due to its high prevalence, particularly among adolescents, countrywide and worldwide (Mummé, Mildred & Knight, 2017; Miller & Prinstein, 2019). In general adolescents, the lifetime NSSI rate was 17%–23% (Brown & Plener, 2017; Gillies et al., 2018), while in clinical samples of adolescents, the rate is as high as 50% (Plener et al., 2015). In China, the epidemiological survey reported that the prevalence of NSSI ranged from 33.7% to 51% in community adolescents (Hu et al., 2020; Liu et al., 2020), and was on the rise (Zhu et al., 2021). Research revealed that NSSI not only predicted future suicidal ideation and behavior in adolescents (Kiekens et al., 2018) but increased the risk of emotional disorders, such as depression and anxiety (Schatten, Andover & Armey, 2015). Numerous studies consistently confirmed a high correlation and comorbidity between depression, anxiety, and NSSI (Jiao et al., 2022; Wang & Liu, 2019; Wester, Trepal & King, 2018). Recognizing NSSI, along with depression and anxiety, as high-priority psychological crises in adolescents (Richard & Burl, 2017), researchers emphasized the comprehension of the complex processes associated with these phenomena.

Various theories interpreted the relations between NSSI and depression and anxiety. The Two-Factor Structure model (Klonsky et al., 2015) explained that both interpersonal (communicating distress, influencing others, seeking support) and intrapersonal functions (emotion-regulation, avoidance of aversive affective, self-punishment) might underlie acts of NSSI. In addition to this, two meta-analysis studies identified the regulation of distressing emotions as a key function of NSSI (Klonsky, 2007; Taylor et al., 2018). Namely, individuals who engaged in NSSI were to escape or refrain from negative emotional experiences, thereby experiencing temporary psychological relief and relaxation (Chapman, Gratz & Brown, 2006; Nock, 2010). Adolescent self-injurers in a community sample, where 80% indicated they self-injured because “I felt very unhappy or depressed” and 45% endorsed the reason “It helped me to release tension or stress and relax” (Laye-Gindhu & Schonert-Reichl, 2005). In empirical studies, several researchers confirmed a significant relief from high-arousal negative emotions immediately following NSSI behaviors, suggesting that NSSI could be an efficient and effective method of regulating emotion (Claes et al., 2010; Kranzler et al., 2018; Sim et al., 2009). Recently, Zhong et al. (2018) also found that heroin-dependent patients in China might use NSSI as a coping mechanism to relieve depression, stress, and other negative emotions. Thus, NSSI predicted a reduction in subsequent depression and anxiety.

From a comorbidity perspective, the theory indicated that NSSI frequently co-occurred with depression and anxiety (Middeldorp et al., 2005). Yen et al. (2016) found that in a clinical adolescent sample, chronic depression increased the likelihood of continuous NSSI. Besides depression, anxiety also positively predicted persistent NSSI during the four-year follow-up period (Steine et al., 2020). In addition, NSSI might also be a precursor variable that exacerbated mood deterioration and led to depression or anxiety. For example, Klonsky (2009) discovered that individuals practising NSSI often reported subsequent feelings of shame, guilt, and self-directed anger. These emotions, in turn, may potentiate or exacerbate depressive and anxiety symptoms (Burke et al., 2019; Kim, Thibodeau & Jorgensen, 2011). Thus, NSSI predicted the degree of depression and anxiety, and vice versa.

To summarize, existing literature on the correlation between NSSI and depression and anxiety presented inconsistent findings. Traditionally, scholars explored these relationships using reflective latent variable models, examining items reflecting the manifestation of an underlying psychological construct (Fried, 2017). However, the network theory of psychopathology (McNally, 2016) challenged this approach, positing comorbidity as an intrinsic feature (interactions between symptoms) of mental disorders rather than an artifact of the diagnostic system due to overlapping criteria. According to network analysis, psychiatric disorders were a system where symptoms could either reinforce or inhibit each other (Borsboom, 2017). In this system, highly “central” symptoms, which have the strongest connections to other symptoms, played a key role in spreading symptom activation throughout the network (Borsboom, 2017). Additionally, “bridge” symptoms facilitated the spread of activation from one disorder to another, establishing a vicious cycle that triggered the formation and continuation of comorbid conditions (Cramer et al., 2010). Therefore, network analyses can quantify and visually display the relations between NSSI and depression and anxiety at the symptom level. Moreover, identifying central and bridging symptoms within these networks can deepen our insight into the mechanisms underlying comorbid disorders and inform the development of effective intervention strategies (Fried & Cramer, 2017).

Recently, a few studies employed a network analysis approach to decipher the complex interconnections within the network structure of NSSI and depression and anxiety. Zhou et al. (2023) found that the severity of depressive symptoms was the sole factor linked to both NSSI and suicide in Chinese college students. Duncan-Plummer et al. (2023) pointed out that the students with a past of NSSI had trouble managing extremely intense and undesired negative emotions through network comparison tests. Additionally, a study by Buelens et al. (2020) identified loneliness, anxiety, and negative effects as antecedent variables of NSSI at the symptom level.

However, prior studies had no direct proof of the network relations between NSSI and depression and anxiety at the symptom level. They were only based on cross-sectional data, which identified the correlations between symptoms, yet did not account for dynamics among variables through lagged predictions. For instance, as cross-sectional network analyses lack directionality, researchers struggled to deduce whether the symptoms with centrality were the triggers to other symptoms or triggered by others. The cross-lagged panel model (CLPM) facilitated researchers in discerning the direction of causality between two or more related constructs (Rhemtulla, Van Bork & Cramer, 2022). By regressing the set of variables at each occasion on the set of variables at the previous occasion, one can estimate the “cross-lagged” effect of each variable on the other over a particular time lag, while controlling for the auto-regressive effect of each variable on itself (Hamaker, Kuipers & Grasman, 2015). Then Rhemtulla, Van Bork & Cramer (2022) combine these two approaches to form a cross-lagged panel network (CLPN). The approach controlled for the autoregressive effects of individual items while examining within-time point (undirected) and across-time point (directed) associations, thereby gauging how a specific item under one construct influences all others under different constructs in the subsequent time frame. This coincided with the temporal connections between symptoms proposed by network theory. Accordingly, it is imperative to scrutinize the temporal and directional connections among symptoms across variables in longitudinal data with the cross-lagged panel network (CLPN) model. Subsequently, it secured relatively potent inferences concerning symptoms.

Additionally, gender differences are also notable to consider when investigating the connection between NSSI and depression and anxiety at the symptom level. Some researchers suggested that the rate of NSSI among girls surpasses that of boys (e.g., He et al., 2023; Xavier, Cunha & Pinto-Gouveia, 2018), whereas one study reported a higher rate in boys than girls (Barrocas et al., 2015). Other studies reported no significant gender differences in NSSI (e.g., Wan et al., 2015; Klonsky, 2011). Moreover, research involving adolescents also uncovered variations in the methods of NSSI between genders. Barrocas et al. (2012) reported that girls are more prone to engage in cutting and carving their skin, whereas boys are more inclined to NSSI through hitting themselves. Considering the potential gender-related differences with a variable-centered approach, it is necessary to use a within-person analysis of the symptoms to understand and address the complex connection between NSSI and depression and anxiety in adolescents.

In summary, the current study utilized the cross-lagged panel network (CLPN) to analyze how individual symptoms of NSSI and depression and anxiety are linked longitudinally and compare boys’ and girls’ symptom-level network structures. Given the exploratory nature of this study rather than confirmatory and the scarce evidence from prior studies, the study was not built on explicit hypotheses but on a set of research questions derived from the comprehensive literature review. The first question delved into the central symptoms of NSSI and depression and anxiety. Identifying these symptoms helped investigate which symptom played the most significant role in triggering and sustaining the psychopathology network of NSSI-depression-anxiety as a whole. The second question focused on what bridging symptoms existed between NSSI and depression and anxiety within the specific contexts of adolescence. These symptoms could show the longitudinal relation at the symptom among them. The third question looked into the existence of sex differences at symptom-level network structures. By exploring the three questions, we aspire to disclose the specific symptoms that act as pivotal connectors over time and the subtle ways these connections may change. This gave a more profound understanding of the intricate relationships between NSSI and depression and anxiety and their gender differences.

Materials and Methods

Participants

Convenience sampling was employed for participant selection in this study. Two public senior high schools in Jinchang City, Gansu Province, China were randomly chosen with the assistance of local education authorities. The procedure was that each school was first numbered (anonymously) by the local education authority, and the experimenter randomly selected two schools from the school numbers according to a table of random numbers and then administered the test to the entire cohort of K-10 and K-11 students from these two schools. In total, 900 eligible participants (414 boys; 486 girls; M age = 15.22, SD = 0.45) were invited to take part in the study. Participants who were reported by their parents to have brain disorders (such as seizures) and psychiatric disorders, or who were receiving psychiatric treatment, would be excluded from the study.

The research program was approved by the Ethics Board of the Northwest Normal University of China (ERB No. 2023093) on January 10, 2023, and conducted in accordance with the World Medical Association’s Code of Ethics (Declaration of Helsinki). The adolescent participants filled out the scale immediately after written consent was obtained from both them and their parents on the site.

Designed and procedure

This longitudinal study used a self-report questionnaire to collect data. It was followed up on two occasions, separated by 6 months. The first measurement was conducted in March 2023 for NSSI, depression and anxiety. The second measure with the same elements was taken in September 2023. The whole process of the questionnaire survey was administered by trained psychological postgraduates in regular classroom setting. Clear and standardized instructions explaining the content, process and confidentiality of the study were given to the students by the assistant. Students could have as much time as necessary for the completion of the measures. They can stop answering at any time if they feel uncomfortable.

Measures

Non-suicidal self-injury

The Adolescent Non-Suicidal Self-Injury Assessment Questionnaire, developed by Wan et al. (2018), was used to assess adolescent non-suicidal self-injury behaviour. The authors gained permission to use this questionnaire. The measurement was widely employed in Chinese adolescents (Fang et al., 2021; Wan et al., 2019). A total of 12 behaviors (e.g., hitting yourself, pulling your hair, etc.) were examined. For example, one item is “In the last month, have you ever engaged in the following behaviors deliberately harming yourself, but not intended to take your life?”. Items regarding their behaviour frequency were rated from 0 (never) to 4 (always), with higher scores indicating greater frequency of NSSI. The Cronbach’s alpha coefficients for the questionnaire were 0.93 (T1) and 0.90 (T2).

Depression

The 18-item Brief Symptom Inventory covered a list of symptoms (BSI-18; Derogatis, 2001). Several studies demonstrated adequate convergent validity and reliability of the BSI-18 to assess distress in China (Geng et al., 2022) and other contexts (Marco et al., 2018). For the current study, however, we only used a 6-item depression subscale to assess the depression symptoms of adolescents. Respondents were asked to indicate how much the symptoms bothered them in the past 30 days (e.g., “In the past 30 days, have you felt lonely?”) and rated from 0 (not at all) to 4 (extremely). A higher score indicated a higher risk of depression. The Cronbach’s alpha coefficients for this scale were 0.83 (T1) and 0.85 (T2).

Anxiety

For this study, we only used the 6-item anxiety subscale of the BSI-18 to assess the anxiety symptoms of adolescents. Respondents were asked to indicate how much the symptoms bothered them in the past 30 days (e.g., “In the past 30 days, have you felt nervousness?”) and rated from 0 (not at all) to 4 (extremely). A higher score indicated a higher risk of anxiety. The Cronbach’s alpha coefficients for this scale were 0.89 (T1) and 0.91 (T2).

Statistical analysis

Based on the guidelines of CLPN analysis, current networks were estimated in R 4.2.0 to construct the CLPN (Rhemtulla, Van Bork & Cramer, 2022). The total sampling of CLPN was first assessed through an array of nodewise regression models to calculate the effects of autoregressive and cross-lagged while controlling all T1 symptoms and the T1 covariates sex.

Current networks utilize the Glmnet package in R to perform the regularized regression (Friedman, Hastie & Tibshirani, 2010). The regression coefficients (i.e., edge weights) were estimated, with red edges denoted negative relationships, green edges representing positive relationships, and wider and more saturated edges indicating stronger relationships. The Qgraph package was used for network plots and determines node placement utilizing an algorithm that positioned nodes sharing stronger connections (Epskamp et al., 2012). To facilitate visual comparison between networks, the layout and maximum edge weights were uniform across all network graphs.

Furthermore, the study performed separate calculations to assess directionality, with cross-lagged out-EI (summing outgoing edges connected to a symptom) and in-EI (summing incoming edges connected to a symptom). Subsequently, the study gauged the accuracy and stability of edge weights by applying bootstrapping from the R package bootnet (Epskamp, Borsboom & Fried, 2018). Edge weight accuracy was evaluated in this study by estimating 95% confidence intervals (CIs) around each weight value through a non-parametric bootstrap approach with 1,000 iterations.

Finally, the study used CLPN to estimate separately across genders. This analysis took into account the correlation between edge lists, cumulative overlapping associations, and the Jaccard Similarity Index for two networks with identical edge characteristics.

Results

Sample characteristics and prevalence of NSSI

A total of 900 students were identified, of which 16 were excluded due to leave of absence, suspension, regularity of responses and incomplete responses. Finally, 884 students were included in the analysis, with a response rate of 98.22%. The mean age of the respondents was 15.19 years (SD = 0.48) and 45.58% were boys.

The percentage of participants reporting NSSI was 23.30%. The rate of engaging in NSSI was significant between boys and girls (Meanboys = 0.93, Meangirls = 1.45; p =0.04). Table 1 displayed the means, standard deviations and correlations of all examined variables at each wave. The result showed that NSSI was positively associated with both depression (p < 0.01) and anxiety (p < 0.01).

Table 1 Descriptive statistics and correlations for the main variables.

Variables	M(SD)	1	2	3	4	5	6	
1. Gender								
2. T1NSSI	1.21(3.97)	0.06						
3. T2NSSI	0.93(2.74)	0.06	0.46**					
4. T1Depression	3.19(3.68)	0.13**	0.36**	0.28**				
5. T2Depression	3.19(3.86)	0.15**	0.33**	0.40**	0.55**			
6. T1Anxiety	3.24(4.12)	0.10**	0.37**	0.30**	0.74**	0.53**		
7. T2Anxiety	2.93(4.27)	0.16**	0.34**	0.36**	0.48**	0.83**	0.58**	
Notes.

∗p < 0.05; ∗∗p < 0.01; ∗∗∗p < 0.01.

CLPN results

The CLPN for total participants was visualized as a directed network (refer to online supplemental materials of Fig. S1). In this figure, arrows indicated pairwise temporal relationships between symptoms, taking into account T1 symptoms and the covariates. Details of edge weights can be found in the online supplementary material of Table S1. The autoregressive edges (β = 0.17) exhibited greater strength in comparison to the cross-lagged edges (β = 0.01, see Fig. S2 in the online supplementary material). Moreover, the analysis identified a total of 552 assessed cross-lagged edges (255 [28.08%] with β >1).

Afterwards, the network model focused on gender differences. In the gender-specific networks, the coefficient similarity of 0.17 highlighted dissimilar connections between the networks for boys and girls. Specifically, only 55.55% of edges present in the girls’ network overlap with those in the boy’s network. Furthermore, the Jaccard Similarity Index yielded a result of 0.38, which means that the majority of edges in the network of the girl and boy don’t have the same orientation. Figure 1 depicted the gender-specific CLPN networks, wherein autoregressive edges (boys: β = 0.12; girls: β = 0.17) have greater strength than cross-lagged edges (boys: β = 0.00; girls: β = 0.02. The figure showed only the predicted paths with β >0.06. Fig. S3 illustrated a moderate bootstrapped confidence interval for the edge weights.

Figure 1 Cross-lagged panel networks of relationships between NSSI and depression and anxiety in boys and girls.

(A) Green edges signify positive relationships whereas red ones signify negative relationships. (B) The arrows connecting symptoms indicate that symptom A at time 1 is a predictor of symptom B at time 2, taking into consideration all other symptoms and covariates. (C) Thicker edges on the graph correspond to higher levels of regressive coefficients, signifying stronger associations.

In the context of the results for boys (refer to Table S2), self-hitting (N5) emerged as the highest-ranked EI-out variable, and feeling scared (A4) was the highest-ranked EI-out variable in relation to depression and anxiety (see Fig. 2). The two most robust bridging associations between depression and anxiety were identified from feeling tension (A6) to feeling blue (D2; β = 0.23), and from feeling scared (A4) to worthlessness (D6; β = 0.14). Additionally, the three strongest bridging connections between NSSI and depression and anxiety were identified as self-cutting (N9) to feeling blue (D2; β = −0.57), feeling tension (A6; β = −0.52) and indifference (D1; β = −0.49).

Figure 2 T1 → T2 network symptom centrality estimates in boys.

In the context of the results for girls (refer to Table S2), self-biting (N6) emerged as the highest-ranked EI-out variable, and feeling scared (A4) was the highest-ranked EI-out variable in relation to depression and anxiety (see Fig. 3). The most robust bridging connections between depression and anxiety were identified from feeling scared (A4) to indifference (D1; β = 0.16), loneliness (D4; β = 0.14), and worthlessness (D6; β = 0.14). Additionally, the three most robust strongest bridging connections between NSSI and depression and anxiety were identified from carving (N12) and rubbing skin (N11) to indifference (D1; β = −0.31, −0.21).

Figure 3 T1 → T2 network symptom centrality estimates in girls.

Discussion

This study represents the inaugural attempt to concurrently explore the longitudinal associations between NSSI and depression and anxiety at the level of symptoms in senior adolescents utilizing a longitudinal network approach and delves into possible gender differences within these relationships. The current study unveiled central and bridging symptoms within the longitudinal network patterns, differentiated by gender. These insights offered valuable information for targeting interventions and preventive measures to address NSSI, depression, and anxiety in senior adolescents.

Specifically, for boys, self-hitting and feeling scared emerged as the highest-ranked EI-out variable, meaning that they most strongly predicted the other symptoms of NSSI and depression and anxiety six months later, after controlling for all other symptoms at T1. Cutting negatively predicted subsequent feelings of sadness, tension and indifference, potentially serving as a bridge connecting NSSI to depression and anxiety. For girls, biting and feeling scared were central symptoms. Carving and skin rubbing negatively predicted subsequent feelings of indifference, serving as bridges linking NSSI to depression or anxiety. These findings provided evidence for our research question that there were gender differences at the network-level relations between NSSI and depression and anxiety. Our results speak to a broader literature examining the link between gender, NSSI and emotion regulation. Research conducted earlier demonstrated that boys tended to resort to hitting or burning themselves as the primary means of NSSI, whereas girls leaned towards methods with blood (especially, cutting, biting and scratching) (Sornberger et al., 2012). In addition, boys were more likely to choose aggressive behavior to deal with stressful events and negative emotions, whereas girls were more likely to use internal coping methods (avoidance) (Nolen-Hoeksema & Aldao, 2011; Paciello et al., 2008). This difference in gender roles is most pronounced in adolescence (Sornberger et al., 2012). There are probably several factors contributing to complex behaviors. To illustrate, gender socialization regarding emotions can influence the emotional experiences of boys and girls, causing variations in the ways emotions are manifested (Bresin & Schoenleber, 2015). One more underlying cause is biological factors, including disparities in genetics and hormones between the sexes, which might result in behavioral differences (Zahn-Waxler, Shirtcliff & Marceau, 2008). Nevertheless, it is inappropriate to infer solid conclusions from these findings, as they were based on a small sample at a symptom level. Thus, forthcoming research should delve deeper into the possible gender differences in how NSSI relates to depression and anxiety.

Despite gender-based differences in symptom level, the symptoms of NSSI at T1 consistently predicted a reduction in certain symptoms of depression and anxiety at T2. One possible explanation for the result was that NSSI may serve as a regulation function to alleviate negative emotions such as sadness, tension and indifference, which represent overall negative affect states (McElroy et al., 2018). This finding was consistent with established models of emotion regulation (Hasking et al., 2017) and corroborated findings from previous studies (Claes et al., 2010; Kranzler et al., 2018; Sim et al., 2009), demonstrating that senior adolescents perceive NSSI as coping strategy to flee from reality and refrain from their negative emotions. Adolescents in high school are in a peak period for NSSI (Plener et al., 2015) due to increased academic pressure, conflict, and increased frequency of depressed and anxious moods (Tang et al., 2019). The challenges of coping with resulting distress, coupled with a propensity to suppress the behavioral expression of emotions, can intensify distress and arousal. In the absence of alternative strategies, this heightened risk may contribute to using NSSI as a means to decrease arousal and modulate emotional responses (Gratz & Roemer, 2004). Therefore, senior adolescents may experience temporary relief through NSSI regardless of potential negative outcomes such as painful and physical consequences. This suggests that NSSI serves not only as a strategy to alter emotions but also as a means to express negative emotions. In this context, NSSI may provide senior adolescents with a sense of control and agency when faced with complex and confusing emotions (Hasking et al., 2017).

An alternative explanation for this result is that NSSI serves a function in interpersonal influence, such as seeking support from others. “The desire for care and attention” was reported as the reason for self-harm by 24% of participants in inpatient research (Herpertz, 1995; Klonsky, 2007). The reason “I want others to see how desperate I am” was endorsed by over 30% of self-injuring adolescents in non-clinical studies (Laye-Gindhu & Schonert-Reichl, 2005). Hence, it can be inferred that senior adolescents who engage in NSSI may diminish depression and anxiety by receiving attention, care, and support from parents, teachers, and other significant others. Previous research has established that support from significant others can accommodate adolescents’ psychological needs, leading to a reduction in negative emotions (Manna et al., 2022; Lyell et al., 2020).

Alongside these outcomes, current network analysis showed that feeling scared was characteristic of anxiety, and emerged as the key bridge symptoms connecting depression and anxiety. The result suggested that symptoms of anxiety were predictors of subsequent symptoms of depression among senior high school students. This pattern is consistent with previous findings at the mean level (Wittchen et al., 2000), which found that anxiety is often the primary condition preceding secondary depression during adolescent development. This suggested that feeling scared exhibited a stronger link to symptoms of depression compared to alternate components of anxiety, thereby exerting a crucial impact on depression and contributing to its development and maintenance. It implies that senior adolescents frequently exhibit anxiety symptoms caused by external stressful events along with mild depression symptoms in the early stages.

To conclude, this study identified the central symptoms and bridge symptoms which provided new insight into the relations between NSSI and depression and anxiety at symptom level. Gender differences in network structure between boys and girls were also found in the study. These gender-specific variations indicate the importance of tailoring interventions to address the unique symptoms of different genders. Furthermore, symptoms of NSSI at T1 were a reliable predictor of a decline in certain symptoms of depression and anxiety at T2. It suggested that NSSI may alleviate or reduce levels of depression and anxiety in senior adolescents on some specific symptoms. It is notable that, although NSSI provides temporary relief from emotional stress, it is essentially avoidant and negative coping. Taking a long-term view, this will merely distance individuals from more viable and efficient coping strategies, fostering a reliance on NSSI. Therefore, when intervening in NSSI behaviors among senior adolescents, emotional regulation should be consistently prioritized, supporting those who engage in NSSI to acquire alternative and adaptive coping mechanisms. For example, DBT or ERGT aim to improve regulation or tolerance of emotions. Concurrently, there is potential to establish a resilient social support system for individuals involved in NSSI, centering on their authentic internal needs and supplying them with resources to handle stress.

Limitations

Although the current study provides insight into the comorbidity between NSSI and depression and anxiety at individual symptom levels in senior adolescents, there are some limitations. First, it is crucial to interpret the findings as generating hypotheses rather than causal confirmation (Liu et al., 2021). Therefore, the study results should be treated with caution. In future studies, qualitative analyses of the relations between NSSI and both depression and anxiety could complement the result of quantitative analyses and frame the findings in specific cultures and contexts. Second, while CLPN can reveal the direction of potential causal relationships, the optimal time lags to capture relationships between symptoms remain unclear. In this study, a six-month time lag was used, but it is important to acknowledge that relationships between symptoms may present themselves at varying intervals. Therefore, researchers should exercise careful consideration when determining the expected time lag, especially in relation to sampling frequency. For instance, if causal effects appear within minutes or hours (e.g., engaging in NSSI → decreases negative emotion), more frequent sampling would be necessary. In addition, two waves of longitudinal data are not sufficient. It is essential to use more waves to validate the stability of the results in future research. Third, this study sample was recruited using convenient sampling, so the generalizability of our findings to other populations is unknown. Replication of these findings in different populations, such as clinical samples, is necessary to confirm their applicability to diverse groups. Lastly, our interpretation of the longitudinal relationship between the NSSI and depression and anxiety were conducted more on the basis of theory without providing direct evidence. Future studies should use various methods, to shed more light on the effect of NSSI functions on the longitudinal relationship between NSSI and depression and anxiety.

Conclusion

The present research employed a longitudinal network model to identify the unique relations between NSSI and depression and anxiety within and across symptom domains and their gender differences. The findings revealed significant gender differences at the network level. For boys, self-hitting and feeling scared emerged as central symptoms. Cutting negatively predicts subsequent feelings of sadness, tension and indifference, potentially serving as a bridge connecting NSSI to depression and anxiety. For girls, biting themselves and feeling scared were central symptoms. Carving and skin rubbing were negatively associated with subsequent feelings of indifference, serving as bridges linking NSSI to depression or anxiety. Additionally, feeling scared emerged as the key bridge symptoms connecting depression and anxiety. Despite gender-based differences, consistent findings indicated that symptoms of NSSI at T1 were a predictor for a decrease in specific symptoms of depression and anxiety at T2. These findings presented a new outlook on the developmental characteristics of symptoms in the relations between NSSI and depression and anxiety across genders.

Supplemental Information

Supplemental Information 1 The original data

Supplemental Information 2 STROBE checklist

Supplemental Information 3 Supplemental Figures and Tables

Supplemental Information 4 Codebook for the data

We sincerely acknowledge the support of Zekai Li, Haixia Chi, Ruiyan Wang and Qian Yang during the revision of the manuscript. The authors would like to thank the two schools’ teachers and all the participants for their contribution to this study.

Additional Information and Declarations

Competing Interests

Author Contributions

Human Ethics

Data Availability

The authors declare there are no competing interests.

Haiyan Zhao conceived and designed the experiments, performed the experiments, analyzed the data, prepared figures and/or tables, and approved the final draft.

Aibao Zhou conceived and designed the experiments, authored or reviewed drafts of the article, and approved the final draft.

The following information was supplied relating to ethical approvals (i.e., approving body and any reference numbers):

Ethical approval for this study was granted by The Ethics Committee of Northwest Normal University (ERB No. 2023093).

The following information was supplied regarding data availability:

The raw measurements are available in the Supplementary File.

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
