# Peer review of "Longitudinal relations between non-suicidal self-injury and both depression and anxiety among senior high school adolescents: a cross-lagged panel network analysis"

_PeerJ, doi:10.7717/peerj.18134_

## Round 0.1 · original submission · Major Revisions

The paper still has a lot of problems, so major revision is needed. I suggest the authors to write the paper according to the STROBE guideline strictly.

Reviewer 1 ·

Basic reporting

Thank you for giving me the opportunity to review this study on the longitudinal association between NSSI and both anxiety and depression. Longitudinal research is always needed in this area. The methods used seem rigorous. I have left a few suggestions below that I hope will help to improve the manuscript.

Introduction:

The introduction I found to be concise and to the point. I only have one minor comment:

“These results have empirically supported across multiple populations, showing that individuals engage in NSSI exhibit heightened levels of anxiety and depression (Wester et al., 2018)”

This statement is not supported by the one reference. I would suggest referencing a meta-analysis or a number of studies with different populations.

Writing:

In some places (particularly in the methods) there is a switch between the tense of the language used. I would suggest using past tense throughout the methods to improve the flow.

Clarity of figures:

Figure 1-3 are quite small and blurry... is it possible to make the image bigger and clearer?

Experimental design

Aims and hypotheses:

The aim of the research is clear; however, the authors make no predictions about the directions/strength of the associations. I understand that the secondary aim is exploratory, but I would expect some predictions to be made with regards to the primary aim.

Validity of the findings

Discussion:

“Therefore, researchers should pay more attention to these specific types of NSSI (biting for girls and cutting for boys) and develop targeted intervention programs”

“there is conjecture that feeling scared shares common feature with anxiety or fearfulness when estimating a network of anxiety symptoms. As such, prioritizing interventions and prevention strategies that address feelings scared should be a primary focus in the treatment of depression and anxiety disorders”

Considering that this paper is a clinical psychology paper, I would expect further discussion of these points. What might these tailored interventions look like? Perhaps a clinical implications section would be good to include in the discussion.

·

Basic reporting

Manuscript I D -
Manuscript Entitle – “Longitudinal relations between non-suicidal self-injury and both 2 depression and anxiety among adolescents: A cross-lagged panel network analysis"

Title & Abstract
• The title is appropriate as expresses the basic theme and focus of the study.
• As far the abstract is concerned background of the study need to validate with citation. Moreover a brief methods need to be added.
• Methods in brief need to be incorporated such as sampling method, scales used.
Introduction
Certainly, this research covers a very crucial concern, namely NSSI among adolescents.
• Language is unambiguous, transparent and appropriate.
• Literature supports are in current context.
• Unfortunately, the background has not been established appropriately to present a comprehensive overview of Non-Suicidal Self-Injury (NSSI). It is inferred that the background information should highlight that NSSI is a significant global public health concern due to its high prevalence, particularly among adolescents, both worldwide and countrywide. As this crucial information, specifically statistics on the issue's prevalence worldwide, is missing in the introduction, it needs to be added.

Results
The author did well in describing the results. However, it is suggested to break the result according to objectives which is suggested to add, to make the presentation more simplify, specific and convenience for readers. For instance -
• Among boys, self-hitting and feeling scared emerged as central symptoms. Cutting negatively predicts subsequent feelings of sadness, tension and indifference, potentially serving as a bridge connecting NSSI to depression and anxiety. Additionally, feelings of tension and scared were key bridge symptoms linking depression and anxiety.
• For girls, biting and feeling scared were central symptoms. Carving, skin rubbing, and self-hitting were negatively associated with subsequent feelings of indifference, serving as bridges linking NSSI to depression or anxiety.
• Feeling scared also emerged as a critical bridge symptom between depression and anxiety. Consistently, these findings highlight NSSI as an emotional regulation strategy for adolescents to alleviate depression and anxiety.
I believe that it needs specific focus rather than compile all together and make the thing more complicated for readers. For instance,
• Separate study should focuses on “Gender wise differences in the symptomatology of NSSI, depression, and anxiety”.
• The other study should attempt to explore symptomatology of NSSI, depression, and anxiety among adolescents. Or find out the lLongitudinal relations between non-suicidal self Injury and both depression and anxiety among adolescents regardless of gender difference.
Discussion
The description of results is found to be relevant with the result. However, some suggestions can be consider to improve such as-
The literature support was established with outdated studies 2020 and before in discussion. Current literature, particularly studies from 2020 or later, provides more relevant and up-to-date support.
Author is suggested to consider Alternative Explanations, as not seems touching on theoretical ground. Hence, suggested to briefly touch on alternative explanations for the observed relationships. This demonstrates a balanced perspective and shows that you've considered different interpretations of the data.
Future Directions need to be added which speculated to give potential avenues for future research. What questions remain unanswered? Are there other new variables that could be explored to further our understanding of relationship of NSSI and emotional Disorder specifically, anxiety and depression.
Conclusion
The conclusion seems well described and more precise.
Figures & Tables
Authors need to check and correct or update the “Fig.2. T1 ³ T2 network symptom centrality estimates in girls”. As this figure is seems for Boys not for girls and subsequent figure (Fig.3.) is for girls.
References
The reference style not adheres to APA 7th edition, as doi and url is missing.
Overall Comment –
Recommended for publication with Minor corrections by working on suggestions.

Experimental design

Methods

However, the purpose or aims of the study are mentioned at the end of introduction. But author is suggested to frame or add the objectives to make the outline of the study more structured.

The use of convenience sampling limits the generalization and external validity of the present findings. The author should endeavor to expand the research in the future to overcome this limitation.
Apart from these points, the rest of the methodological aspects have been appropriately addressed.

The design and statistical analysis methods are align with the objectives appropriately.

Validity of the findings

The use of convenience sampling limits the generalization and external validity of the present findings.

The author should endeavor to expand the research in the future to overcome this limitation.

Additional comments

Overall Comment –
Recommended for publication with Minor corrections by working on suggestions. Specifically-

Figures & Tables
Authors need to check and correct or update the “Fig.2. T1 ³ T2 network symptom centrality estimates in girls”. As this figure is seems for Boys not for girls and subsequent figure (Fig.3.) is for girls.

References
The reference style not adheres to APA 7th edition, as doi and url is missing.

Reviewer 3 ·

Basic reporting

This is an interesting article on the symptom-level longitudinal associations between depressive and anxiety symptoms and NSSI behaviors. The main strength of this study is the use of a cross-lagged effect model with psychological network analysis. However, the paper still needs substantial revisions.



First, the abstract should quantify the findings using main statistics and accurate P values. It is not adequate to describe the results without any supporting data.



Second, in the title, the term “adolescents” should not be used because the authors only investigated 11-12th-grade senior middle school students. In fact, the age range for adolescents is 12-29 years.



Third, I disagree with the current conclusion that NSSI played an emotional regulation strategy unless the authors provide direct data on emotional regulation strategy. The significant symptom-level association between depression and anxiety and NSSI cannot conclude such a conclusion. The authors need to assess the function of NSSI in this study.The corresponding discussion should als be revised.


Fourth, in the introduction of the main text, the authors reviewed the emotional regulation strategy of NSSI. However, this is only one of the many functions of NSSI. A comprehensive review of the subtype and function of NSSI is needed (i.e., PMID: 29957566, PMID: 33354431). In particular, the current data cannot support the emotional regulation role. In this part, please explain why BPD is. I suggest the authors review the sex difference in NSSI behaviors because this is one of the focuses of this study. Also, a more comprehensive review of the cross-lagged model is needed (i.e., PMID: 27013536). A basic question is that two visits of longitudinal data are not sufficient for a cross-lagged model because there is no sufficient data to support the stability of the proposed cross-lagged paths. In general, three visits are required.



Fifth, the study city should be clearly indicted in the methodology. Please further explain whether verbal informed consent from students and their parents is appropriate for this study, considering that the middle school students are not adults. The Adolescent Non-Suicidal Self-Injury Assessment Questionnaire has two parts, but the authors did not use the function part. This part is important for answering the current research question. My further question is that the first part is mainly an assessment of 12 NSSI behaviors, so “with higher scores indicating greater severity of NSSI” is misleading. The severity of NSSI is not dependent on its frequency. The NSSI-induced medical treatment and the severity of physical injury is the severity of NSSI. Please indicate the reliability and validity of BSI-18 in the Chinese population. In statistics, network comparisons between male and female students should be performed to assess the sex difference in the network (i.e., PMID: 37705850, PMID: 38797390).



Sixth, the authors need to carefully check the reference list since I did not find Wan et al. (2018) Yu et al., 2023. In line 432, the cited paper is incomplete.



Seventh, the English language of this paper needs substantial revisions before resubmission.

Experimental design

.

Validity of the findings

.

---

## Round 0.2 · accepted · Accept

I am pleased to accept this revised paper.

Reviewer 1 ·

Basic reporting

no comment

Experimental design

no comment

Validity of the findings

no comment

Additional comments

After reviewing the response to the reviewers' document and edited manuscript, I believe the authors have answered all points raised by the reviewers.

Reviewer 3 ·

Basic reporting

I have no further comments.

Experimental design

I have no further comments.

Validity of the findings

I have no further comments.

Additional comments

I have no further comments.